# Pharmacoepidemiological Analysis of Antibacterial Agents Used in a Provisional Hospital in Aktobe, Kazakhstan, in the Context of COVID-19: A Comparison with the Pre-Pandemic Period

**DOI:** 10.3390/antibiotics12111596

**Published:** 2023-11-06

**Authors:** Aigerim A. Balapasheva, Gaziza A. Smagulova, Aigul Z. Mussina, Liliya E. Ziganshina, Zhansulu Zh. Nurgaliyeva

**Affiliations:** 1Department of Pharmacology, West Kazakhstan Marat Ospanov Medical University, Aktobe 030012, Kazakhstan; g.smagulova@zkmu.kz (G.A.S.); a.mussina@zkmu.kz (A.Z.M.); pharma@zkmu.kz (Z.Z.N.); 2Russian Medical Academy of Continuous Professional Education, 125993 Moscow, Russia; 3Department of General and Clinical Pharmacology, Peoples’ Friendship University of Russia Named after Patrice Lumumba (RUDN University Named after Patrice Lumumba), 6 Miklukho-Maklaya St., 117198 Moscow, Russia; 4Department of Pharmacology, Kazan State Medical University, 49 Butlerov Street, 420012 Kazan, Russia

**Keywords:** pharmacoepidemiological studies, COVID-19, antibiotic resistance, ATC/DDD methodology, antibacterial agents

## Abstract

In the context of the global spread of Coronavirus Disease 2019 (COVID-19), the issue of evaluating and optimizing the use of antibacterial drugs becomes especially relevant. The coronavirus pandemic has provided a unique opportunity to study the dynamics of the consumption of antibacterial agents and their impact on public health. The rational use of antibiotics is a key aspect of the fight against antimicrobial resistance, which makes this study particularly important. The aim of this study was to assess changes in the consumption of antibacterial drugs among patients hospitalized with COVID-19 during the peak of the 2020 pandemic and compare them with data from 2019 prior to the pandemic. This study collated data on antibacterial drug consumption in a regional hospital in Aktobe, which served a large population of patients during the pandemic. A pharmacoepidemiological study was conducted using the Anatomical Therapeutic Chemical (ATC)/Defined Daily Dose (DDD) methodology. The pharmacoepidemiological study using the international ATC/DDD methodology revealed a concerning pattern of irrational consumption of antibacterial drugs, including cephalosporins, azalides, second-generation fluoroquinolones, and systemic aminoglycosides in Aktobe. Among antibacterial drugs during the pandemic, the most significant increase in consumption was from the group of cephalosporins (19,043 DDD/100 bed-days). The share of their consumption was 35.4% of the total consumption of antibacterial drugs. Pharmacoepidemiological studies using the international methodology ATC/DDD showed an alarming picture of irrational consumption of antibacterial drugs of the group of cephalosporins, azalides, fluoroquinolones, and aminoglycosides in Aktobe, and, in this case, excessive use of the identified antibiotics raises concerns about the possibility of increasing the problem of resistance to microbes.

## 1. Introduction

On 30 January 2020, the World Health Organization (WHO) declared the outbreak of severe acute respiratory syndrome coronavirus 2 (SARS-CoV-2) a public health emergency of international concern. In Kazakhstan, the first cases of COVID-19, caused by this coronavirus, were reported on 13 March 2020. At the onset of the pandemic, protocols for diagnosing and treating the infection were frequently updated as new information surfaced [1].

Antibacterial drugs, designed to treat bacterial infections, are widely used in medicine. They effectively eliminate or inhibit the growth of bacteria responsible for infections. However, COVID-19 is caused by the SARS-CoV-2 virus, which was first identified in Wuhan, Hubei Province, China, in late 2019.

Antibacterial drugs are not effective against viruses, including SARS-CoV-2. Thus, antibacterial therapy is only prescribed to COVID-19 patients when there are clear signs of a bacterial infection. For instance, antibiotics might be given to patients who have pneumonia as a result of COVID-19, but this is based on general guidelines: the severity of the patient’s condition, risk factors for infections by resistant bacteria, and etiological diagnosis results are all taken into account. The utilization of antibacterial drugs during the COVID-19 pandemic had several motivations:

Combatting secondary infections: COVID-19 patients were at a heightened risk of contracting secondary bacterial infections, particularly in hospital environments. Antibacterial medications, like azithromycin, were used either as a treatment or preventive measure for such infections.

Pneumonia: Some individuals with COVID-19 also contracted bacterial pneumonia. This dual infection necessitated prompt antibacterial intervention [2].

Even before the pandemic’s onset, the irrational use of antibacterial drugs was already a pressing global issue [3,4]. Some systematic reviews have highlighted a disparity between the high rate of antibacterial drug administration to COVID-19 patients and the actual prevalence of concurrent bacterial infections. This discrepancy suggests a potential over-prescription of these drugs. Astonishingly, nearly 78% of COVID-19 patients were prescribed antibiotics. The most commonly administered were cephalosporins, which were prescribed to 30.1% of the patients, closely followed by azithromycin at 26%. Notably, these antibacterial drugs were given to COVID-19 patients irrespective of the disease’s severity [5,6].

At the pandemic’s onset, some European medical institutions even experimented with the third generation of cephalosporin-class antibacterial drugs as a potential treatment for the virus. Such practices, coupled with misdiagnoses leading to incorrect prescriptions or unjustified antibiotic courses, can fuel the rise of antimicrobial resistance [7,8].

Kazakhstan, in alignment with the broader global community, adheres to the WHO’s recommendations. However, even with a slight reduction in the consumption of systemic antimicrobial agents in recent years, the irrational use of antibacterial drugs continues in Kazakhstan [9,10].

According to the WHO’s assessment, seven factors contribute to the irrational use of antibacterial drugs:-Insufficient skills and knowledge of the prescribing physician.-Unethical promotion of pharmaceuticals by drug companies.-Unrestricted access to medications.-Profits derived from drug sales.-The high cost of medications.-An excessive workload for medical personnel.-The absence of coordinated national pharmaceutical policies [11].

Given these challenges, it is crucial to bolster antimicrobial stewardship (AMS) and develop clear antibiotic usage policies [12,13]. The inappropriate prescription of antimicrobial agents to COVID-19 patients poses a significant concern, especially with the escalating risk of antimicrobial resistance (AMR). This resistance can lead to increased morbidity, mortality, and a rise in both healthcare and societal costs [14,15].

Pharmacoepidemiology plays a pivotal role in monitoring antibiotic use in both hospital and community settings. It aids in understanding the factors that influence antibiotic consumption and gauging their effects [16]. The aim of this study was to examine the shifts in antibacterial agent consumption among hospitalized COVID-19 patients during the 2020 pandemic’s apex and compare these findings with 2019, pre-pandemic, data. Another objective was to pinpoint target indicators that would aid in implementing measures for the rational use of antibacterial agents.

## 2. Results

In this study, an increase in the consumption of antibacterial drugs during the pandemic was observed compared to the pre-pandemic period. The total volume of antimicrobial agent consumption in 2020 surpassed that of 2019. Specifically, in 2019, the total volume of antibiotic consumption was 26,188 DDD/100 bed-days, but in 2020, during the height of the COVID-19 pandemic, this number more than doubled, reaching 53,786 DDD/100 bed-days.

Upon analyzing data from late March to December 2020, it was found that a total of 27 unique antibiotics were prescribed. During the pre-pandemic period, from January to December 2019, only 23 antibiotics were utilized. These figures include both oral and parenteral forms of antibiotics. Every antibacterial agent used systemically between 2019 and 2020 was classified under the J01 code, following the WHO’s ATC classification, and the average Defined Daily Dose (DDD) for each medicinal product was determined. Comprehensive results of the antibacterial agent consumption assessment can be found in Table 1.

Among all the consumed antibacterial drugs, the top ten antibiotics used from 2019 to 2020 were identified. Of the antibiotics used during the pandemic, the most notable rise in consumption was seen in the third-generation cephalosporin group. Specifically, ceftriaxone ranked first with a consumption rate of 19,043 DDD/100 bed-days, accounting for 35.4% of the total antibiotic use. During a similar period in 2019, this drug was not consumed as extensively, registering a rate of only 6151 DDD/100 bed-days. Before the pandemic, cefotaxime led the list with a rate of 6151 DDD/100 bed-days, but during the pandemic, its ranking dropped to third place.

During the pandemic, metronidazole ranked second in consumption with a rate of 6.906 DDD/100 bed-days. The consumption of this drug increased by 27.57% during the COVID-19 period, compared with the previous 5.414 DDD/100 bed-days. Additionally, there was a marked rise in the consumption of azithromycin, reaching 3.476 DDD/100 bed-days. Notably, the use of parenteral amoxicillin/clavulanic acid surged, increasing fourfold during the pandemic. In 2020, its consumption reached 2.026 DDD/100 bed-days, compared to only 0.363 DDD/100 bed-days in 2019—a six-fold increase.

Another significant observation is the increase in consumption of the oral form of levofloxacin. In 2019, it stood at 0.177 DDD/100 bed-days, but in 2020, amid the COVID-19 pandemic, it surged to 3.389 DDD/100 bed-days, while the parenteral form was not used at all. Additionally, there was a marked rise in the consumption of gentamicin, an aminoglycoside antibiotic, jumping from 0.003 DDD/100 bed-days to 1.811 DDD/100 bed-days. Another antibacterial drug, the second-generation fluoroquinolone called ciprofloxacin, also showed a notable increase in consumption, reaching 2.872 DDD/100 bed-days in 2020, which is noteworthy because it was not prescribed at all in 2019. Comparative metrics for both periods can be viewed in Figure 1.

Significant changes in antibiotic consumption were observed during the pandemic. These trends suggest a decreased demand for certain antibacterial drugs during this period, potentially reflecting shifts in disease prevalence or alterations in treatment methodologies. For instance, the consumption of cefazolin decreased by 14.36%, cefuroxime by 4.18%, and cefotaxime by 2.44%. On the other hand, the use of metronidazole rose by 27.59%, while levofloxacin witnessed a substantial increase of 194.7%. Such dramatic increases might indicate a rise in the incidence of infectious diseases for which these antibiotics are particularly effective, or they might highlight their inclusion in the combined therapy protocols for COVID-19. These variations in antibiotic usage could be influenced by a range of factors, such as updates in clinical guidelines, shifts in the prevalence of infectious diseases, or the availability of specific drugs.

To perform a statistical analysis to assess the significance of changes in the consumption of the listed drugs during the pre-pandemic and pandemic periods, the t-text for paired samples was used. This test is suitable for comparing the means of two related groups (in this case, the consumption of each drug before and during the pandemic).

-Amikacin 500 mg injection:

difference (during−before) = 0.378 − 0.269 = 0.109

Assuming an arbitrary standard deviation of 0.1 for this example:

t = 0.109/(0.1/sqrt(1)) = 1.09

-Amoxicillin/clavulanate 0.5/0.1 g:

difference = 2.026 − 0.363 = 1.663

Assuming an arbitrary standard deviation of 0.5 for this example:

t = 1.663/(0.5/sqrt(1)) = 3.326

-Azithromycin 500 mg tablet:

difference = 3.476 − 0.024 = 3.452

Assuming an arbitrary standard deviation of 0.5 for this example:

t = 3.452/(0.5/sqrt(1)) = 6.904

Ampicillin sodium salt 500 mg: difference = −0.019, t = −0.19

Vancomycin 1 g: difference = −0.133, t = −1.33

Gentamicin 80 mg ampoule: difference = 1.808, t = 18.08

Doripenem 500 mg: difference = 0.036, t = 0.36

Clarithromycin 500 mg tablet: difference = 0.139, t = 1.39

Levofloxacin 500 mg/100 mL: difference = 2.411, t = 24.11

Meropenem 1000 mg: difference = 0.004, t = 0.04

Metronidazole 500 mg solution: difference = 1.492, t = 14.92

Ofloxacin 200 mg/100 mL: difference = 0.086, t = 0.86

Cefotaxime 1 g: difference = 0.15, t = 1.5

Ceftriaxone 1000 mg: difference = 13.475. t = 134.75

Cefepime 1000 mg: difference = 0.054, t = 0.54

Cefazolin 1000 mg: difference = 0.706, t = 7.06

Cefuroxime 500 mg tablet: difference = 0.029, t = 0.29

Cefuroxime 1500 m: difference = 0.03, t = 0.3

For a *t*-test with df = 1 at alpha = 0.05, we typically use a critical t-value of around 12.71 for a two-tailed test. Any t value larger than this in absolute value is considered significant.

Conclusions:

Significant increases: Amoxicillin/clavulanate, azithromycin, gentamicin, levofloxacin, metronidazole, and ceftriaxone.

Significant decreases: None that exceed the critical t-value.

All other antibiotics did not show statistically significant changes based on this rough estimation.

Based on the presented data, it is evident that the pandemic significantly influenced antibiotic usage. As a result, certain classes of antibiotics experienced heightened demand, while the consumption of others declined. Specifically, cephalosporins emerged as the most frequently used antibiotics during the pandemic, with consumption rates surpassing those seen in the pre-pandemic period. In contrast, both macrolides and carbapenems maintained low consumption levels during both the pandemic and pre-pandemic phases (Figure 2).

## 3. Discussion

The spread and incidence of COVID-19 have varied significantly across different regions of Kazakhstan. Specifically, in the Aktobe region, the peak in case numbers for 2020 was observed in June and July. Among all regions in Kazakhstan, Aktobe ranks 12th in terms of reported COVID-19 cases.

For this study, we collected data on the consumption of antibacterial drugs from a regional hospital in Aktobe, a facility that catered to a vast patient population during the pandemic. This comparative study, focusing on antibacterial drug usage amidst the COVID-19 crisis using the ATC/DDD methodology, is pioneering in the context of Kazakhstan. It is evident from our findings that the COVID-19 pandemic profoundly influenced the prescription patterns of antibacterial drugs, leading to a considerable uptick in consumption. Notably, during this period, antibiotics like ceftriaxone, azithromycin, levofloxacin, amoxicillin/clavulanic acid, and ciprofloxacin emerged as the predominant choices in hospital settings [16]. This surge in antibiotic administration for COVID-19 patients can partly be attributed to prevailing panic. As per existing guidelines, prescribing antibacterial therapy for COVID-19 patients is deemed appropriate only when there are clear indications of a concurrent bacterial infection alongside the coronavirus infection.

The authors sought to juxtapose the study’s results with those from other countries. A comprehensive search was undertaken across databases including Scopus, Web of Science, PubMed, and Google Scholar to identify global research on similar topics. When aligning the findings from this study with international data, both parallels and distinctions emerged. For instance, azithromycin and ceftriaxone were prevalently prescribed during the pandemic across various countries. Studies demonstrated that antibiotic prescriptions during the COVID-19 pandemic in countries such as Jordan, England, and Spain witnessed a decline in DDD numbers [17,18]. In contrast, nations like Pakistan, Egypt, Bangladesh, and Slovenia saw a marked uptick in this metric [19,20]. To illustrate, a retrospective analysis from five hospitals in Pakistan during the COVID-19 outbreak noted an elevation in the consumption of azithromycin, ceftriaxone, and amoxicillin/clavulanic acid by 11.5 DDD/100 bed-days [21,22].

A comparison of 2019 and 2020 data from a hospital in the Republic of Doboj, Serbia, underscored a pronounced surge in azithromycin consumption in 2020. This was also observed for ceftriaxone (at 14.0 DDD/100 bed-days), meropenem (at 2.33 DDD/100 bed-days), and vancomycin (at 1.54 DDD/100 bed-days) [23]. However, in the context of this study, meropenem and vancomycin remained relatively consistent in their consumption patterns.

Azithromycin, a readily accessible antibacterial drug, boasts a generally commendable safety profile. Notably, azithromycin holds the potential to address co-existing infections and secondary bacterial infections that might manifest in patients suffering from respiratory viral conditions [24,25]. However, the pervasive prescription of this antibiotic during the pandemic is a cause for concern. The WHO has also highlighted the undue reliance on azithromycin for treating COVID-19, notwithstanding the lack of formal endorsement [26]. 

In research carried out in Vanuatu, both pre-COVID-19 and amidst the pandemic, penicillin emerged as the predominant class of antibacterial drugs prescribed to in-patients. They represented approximately 70% of all prescriptions across both timeframes. Cloxacillin, within this antibiotic class, stood out as the primary choice, accounting for 37% of the prescriptions in each period. Interestingly, in our study, cloxacillin was not prescribed, a probable explanation being its omission from the Kazakhstan National Formulary (KNF).

A comprehensive retrospective study examining the consumption dynamics of antibacterial drugs across 66 Spanish hospitals during the pandemic revealed trends akin to our findings, especially concerning azithromycin. Its usage escalated from 3.26 DDD/100 bed-days pre-pandemic to 6.69 DDD/100 bed-days during the pandemic.

Conversely, counterparts in Saudi Arabia undertook a comparative analysis of antimicrobial drug consumption for the years 2019 and 2020. They gauged antibiotic consumption in hospitals and expressed it in terms of a fixed daily dose (DDD) per 100 bed days, adhering to the WHO guidelines. The overarching trend for 2020 was a 16.3% uptick in total antimicrobial consumption compared to 2019. Notably, the year 2020 witnessed a drop in the consumption of fourth-generation cephalosporins (−30%), third-generation cephalosporins (−29%), and penicillin combinations (−23%). On the flip side, antibiotics that witnessed heightened consumption in 2020 relative to 2019 encompassed linezolid (374%), vancomycin (66.6%), and carbapenems (7%). Notably, linezolid remains the sole antibiotic from the reserve group cited in the hospital’s pharmacological references. In Aktobe-based hospitals, linezolid is not administered at all, even though it finds a place in the KNF.

In Jordan, in 2019, the most widely administered antibacterial drugs included broad-spectrum penicillin, β-lactamase inhibitors, macrolides, fluoroquinolones, and combinations of penicillin. However, 2020 saw a shift, with β-lactamase inhibitors, macrolides, extended-spectrum penicillin, and fluoroquinolones leading the chart.

Kazakhstan’s antibiotic consumption pattern during the COVID-19 pandemic echoed a global trend: a significant surge in the use of these drugs. In this country, the most pronounced increase was seen with cephalosporins, metronidazole, and azithromycin.

When juxtaposed with data from Jordan and Spain, azithromycin’s elevated consumption becomes evident. Its prolific use, even in the absence of the WHO’s endorsement for treating COVID-19, is intriguing. This trend could be attributed to azithromycin’s potential efficacy against secondary bacterial infections and its demonstrated antiviral activity in vitro. Nevertheless, discrepancies arise when observing the use of other antibiotics. For instance, Kazakhstan recorded a stark ascent in cephalosporin usage, whereas Saudi Arabia noted a decline for the same class of drugs. Such disparities might be rooted in each nation’s treatment protocols, drug accessibility, and prevailing disease patterns. It is also noteworthy that Saudi Arabia saw an uptick in the use of linezolid, vancomycin, and carbapenems. In contrast, Kazakhstan either abstained from using these drugs or maintained a consistent consumption level across the observed periods.

The consumption trends for antibacterial drugs in any given country are shaped by a confluence of factors. These include the severity of COVID-19’s spread, nationally recommended treatment modalities, drug availability, and the unique structure of each nation’s healthcare system. Contextualizing this study’s findings within the global data underscores the imperative for ongoing analysis and surveillance of antibiotic usage. Such measures are pivotal to staving off drug resistance and streamlining their therapeutic applications.

Recommendations and information regarding COVID-19 were continually updated, making it challenging to ensure that the latest information was readily accessible to all medical professionals and patients. The availability of certain antibacterial drugs was expanded during the pandemic in response to the heightened demand. This influenced prescription decisions, especially in situations of urgent need. Doctors relied on certain classes of antibacterial drugs due to their proven effectiveness in past clinical practice. This trust was likely bolstered by the fears and uncertainties surrounding a new and not yet fully understood virus.

## 4. Materials and Methods

### 4.1. The Study Design and Research Period

This study was conducted as part of a scientific research project at Marat Ospanov West Kazakhstan Medical University. The project is titled ‘Concomitant Bacterial Infections and Pharmacoepidemiology of Antibiotic Resistance in Patients with COVID-19: The Situation in the Aktobe Region’. Ethical approval was secured from the local bioethics committee (Approval No. 8, dated 15 October 2021). The research employed a retrospective, comparative, and descriptive pharmacoepidemiological design to examine the consumption of antibacterial drugs.

Materials used in the research comprised in-patient records, data on the number of bed-days for the years 2019–2020, and pharmacy records detailing antibiotic usage in the hospital, with a significant number of cases occurring in June and July. The authors included all patients who underwent antibacterial treatment in the hospital both prior to and during the pandemic.

### 4.2. The Structure of the Hospital

Prior to the pandemic, the dispensary hospital in this study was a multidisciplinary medical institution in the Aktobe region, boasting a capacity of 532 beds. However, on 16 March 2020, following the declaration of a state of emergency, the hospital began operating under quarantine as per the directive of the Regional Health Department of the Aktobe region, which was in alignment with the Republic of Kazakhstan’s sanitary and epidemiological surveillance authorities’ guidelines for the COVID-19 situation. Subsequently, in line with Order No. 68-5, dated 16 April 2020, the multidisciplinary hospital was repurposed to a 400-bed dispensary hospital dedicated to treating patients with severe pneumonia.

In 2020, the makeshift multi-profile hospital of the Aktobe Medical Center provided in-patient treatment to 2223 patients diagnosed with severe pneumonia and various concurrent diseases. This treatment followed the diagnosis and treatment protocols set by the Ministry of Health of the Republic of Kazakhstan.

Before the onset of the pandemic, the hospital offered services across 25 specialties, encompassing both clinical and paraclinical departments. This included surgical specialties such as neurosurgery, otolaryngology, elective surgery, general surgery, urology, maternity pregnancy pathology, traumatology and orthopedics, vascular surgery, thoracic surgery, an operating room and anesthesia center, an endocrinology department, a respiratory medicine and allergology center, newborn pathology, a neonatal intensive care unit, a gynecological department, and an intensive care unit.

### 4.3. The Methodology Employed and the Statistical Analysis

This pharmacoepidemiological study employed the ATC/DDD (Anatomical Therapeutic Chemical/Defined Daily Dose) methodology, adhering to indices denoting the quantity of DDD/100 bed-days, as currently advocated by the WHO.

The WHO has instituted a suite of tools for drug utilization research to aid in the scrutiny and assessment of prescribing, dispensing, and consuming medication [27]. At present, the ATC/DDD methodology is acknowledged as the “gold standard” for pharmacoepidemiological research on drug usage. The WHO endorses this methodology as an efficient instrument for pharmaceutical statistics and promotes it as the international standard for appraising medication utilization, ensuring both optimal results and quality usage [28].

Historically, the majority of pioneering pharmacoepidemiological studies on drug consumption originated from European and North American countries, leveraging administrative databases established due to their UHC (Universal Health Coverage). In the Southeast Asia Region (SEARO) of the WHO, which accounts for roughly a quarter of the global population [29], a mere one in five drug utilization studies employed the WHO ATC/DDD system, as identified by a systematic review in the area. To our understanding, systematic evaluations for other territories, encompassing the CIS, remain unexplored. Conclusions drawn from this study emphasize that medication utilization research and adoption of the ATC/DDD system should be actively promoted and undertaken in low- to middle-income nations [30]. Typically, DDD values are calibrated based on adult consumption. For drugs greenlighted for pediatric administration, dosage guidelines diverge based on age and body mass. Additional nuances arise when contemplating drug utilization studies involving children.

The ATC/DDD methodology serves as an international language for studying medication consumption with the aim of improving prescription practices, i.e., ensuring their rational use [31].

DDD per 100 bed-days can be applied when considering the use of medications by in-patients. The study drugs are classified under the ATC code J01, which denotes antibacterial agents for systemic use. For each antibiotic, the ATC/DDD index (Anatomical Therapeutic Chemical/Defined Daily Dose) was retrieved from the following website: https://www.whocc.no/atc_ddd_index/, accessed on 10 July 2023. 

The calculation of consumption was performed using the following formula:(1)$$\mathrm{DDD}/100\;\mathrm{bed-days}=\frac{\mathrm{DDDS}}{\mathrm{Total}\;\mathrm{bed-day}}\times 100$$

The DDD represents the estimated average maintenance dose of a medication used for its primary indication in adult patients, serving as a technical unit of measurement. Calculating the intensity of antibiotic use with the ATC/DDD index is not influenced by price or package size, ensuring a consistent DDD for each antibiotic. The WHO determines the DDD value in grams for each medication and updates it annually.

Descriptive statistical methods were employed for data description and analysis. All statistical evaluations were performed using IBM SPSS Statistics version 24.0 for Windows (IBM Corp., Armonk, New York, NY, USA). 

## 5. Conclusions

This pharmacoepidemiological study, employing the international ATC/DDD methodology, uncovered a worrisome trend of an irrational consumption of antibacterial drugs in the Aktobe hospital. This included cephalosporins, azalides, second-generation fluoroquinolones, and systemic aminoglycosides. The excessive consumption of these antibacterial drugs heightens concerns regarding the potential aggravation of microbial resistance. These research findings suggest that the ATC/DDD methodology provides reliable data when assessing antimicrobial drug use within the hospital setting. A study of antibacterial drug consumption among patients hospitalized with COVID-19 in Kazakhstan highlighted a significant surge in the utilization of these medications in 2020 compared to the pre-pandemic period in 2019. Specifically, ceftriaxone consumption increased over threefold, reaching 19.043 DDD/100 bed-days, while azithromycin rose from 0.024 DDD/100 bed-days to 3.476 DDD/100 bed-days. This pattern demands close scrutiny given the potential repercussions. Unwarranted and excessive use of antibacterial drugs can foster microbial resistance, undermining the efficacy of these drugs in future clinical applications. In light of these findings, there is an urgent need to prioritize the development and implementation of strategies promoting the rational use of antibacterial drugs.

This involves enhancing the awareness of both medical professionals and patients, ensuring the judicious availability of antibiotics, and prescribing them based solely on stringent medical indications. These findings mirror the global trend of escalating antibacterial drug consumption during the COVID-19 pandemic, a trend also documented in Spain, Jordan, and other nations. In Kazakhstan, the surge can be attributed to factors such as limited awareness, drug availability, and the trust both doctors and patients place in certain groups of antibacterial drugs. The authors assert that to address these issues, there should be active involvement of clinical pharmacologists in guiding antibiotic use in hospitals. Additionally, creating tailored guidelines for the rational use of antibacterial drugs specific to each hospital and optimizing the utility of microbiological laboratories are imperative steps forward.

## Figures and Tables

**Figure 1 antibiotics-12-01596-f001:**
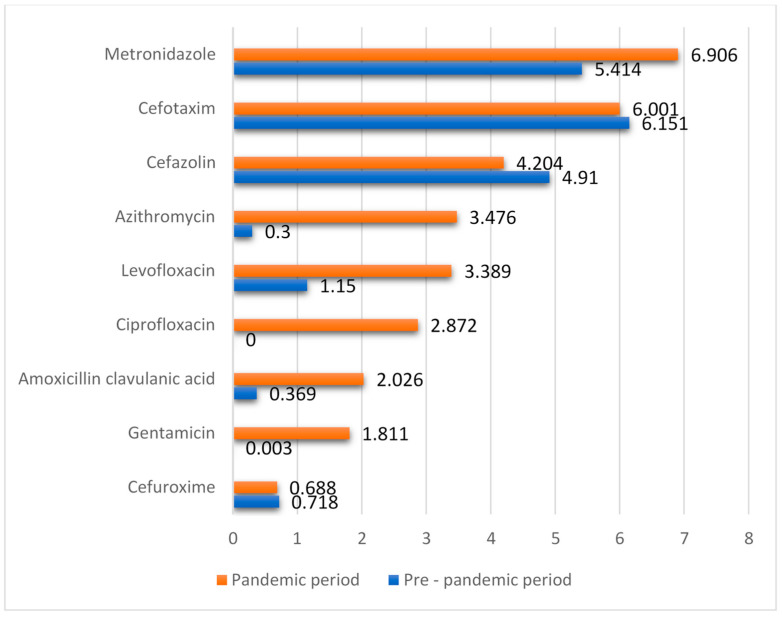
The consumption of the top ten antibacterial drugs during the pre-pandemic period and in the context of COVID-19.

**Figure 2 antibiotics-12-01596-f002:**
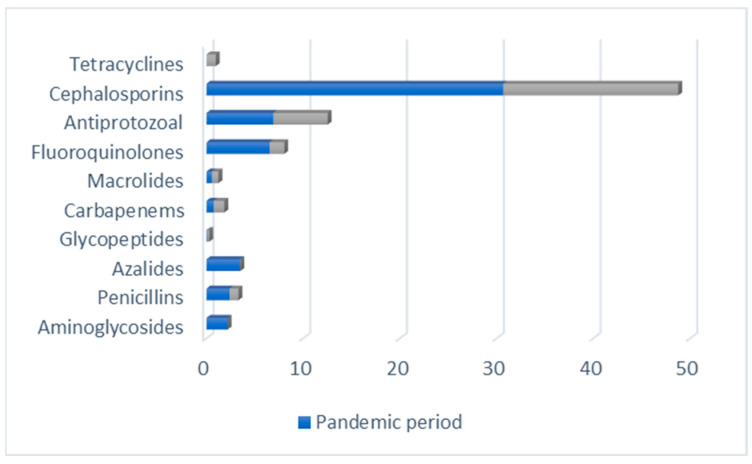
Comparative indicator of antibacterial drugs classes used in 2019–2020.

**Table 1 antibiotics-12-01596-t001:** ATC/DDD analysis of the overall consumption of antibacterial agents in the hospital in 2019–2020.

Prescribed Antibiotics (Injection, Capsule, Tablet)	ATC Code	DDD in the Hospital Per 100 Bed-Days			
		Before the Pandemic: 2019	%	During the Pandemic: 2020	%
Amikacin 500 mg injection	J01GB06	0.269	1.05	0.378	1.48
Amoxicillin 500 mg capsule	J01CA04	-	-	0.018	
Amoxicillin/clavulanate 0.5/0.1 g	J01CR02	0.363	1.43	2.026	7.98
Azithromycin 500 mg tablet	J01FA10	0.024	0.09	3.476	13.03
Ampicillin sodium salt 500 mg	J01CA01	0.200	0.79	0.181	0.71
Vancomycin 1 g	J01XA01	0.210	0.83	0.077	0.30
Gentamicin 80 mg ampoule	J01GB03	0.003	0.01	1.811	6.04
Doripenem 500 mg	J01DH04	0.043	0.17	0.079	0.31
Imipenem + Cilastatin 500 mg	J01DH51	-	-	0.003	
Clarithromycin 500 mg tablet	J01FA09	0.021	0.08	0.160	0.64
Clarithromycin 500 mg vial	J01FA09	-	-	0.343	
Levofloxacin 500 mg/100 mL	J01MA12	0.978	3.85	3.389	13.34
Levofloxacin 500 mg tablet	J01MA12	0.172	0.68	-	
Meropenem 1000 mg	J01DH02	0.400	1.57	0.404	1.59
Metronidazole 500 mg solution	J01XD01	5.414	21.30	6.906	27.17
Moxifloxacin 400 mg/250 mL	J01MA14	0.003	0.01	0.023	0.08
Thiamphenicol 500 mg	J01BA02	-	-	0.005	
Piperacillin 4.5 g	J01CR05	-	-	0.165	
Ofloxacin 200 mg/100 mL	J01MA01	0.160	0.63	0.246	0.97
Cefotaxime 1 g	J01DD01	6.001	23.61	6.151	24.20
Ceftriaxone 1000 mg	J01DD04	5.568	21.90	19.043	74.89
Cefepime 1000 mg	J01DE01	0.499	1.96	0.553	2.09
Cefazolin 1000 mg	J01DB04	4.204	16.54	4.910	19.32
Ciprofloxacin 200 mg/100 mL	J01MA02	-	-	2.872	
Cefuroxime 500 mg tablet	J01DC02	0.195	0.77	0.224	0.88
Cefuroxime 1500 m	J01DC02	0.688	2.71	0.718	2.83

ATC = Anatomical Therapeutic Chemical Classification; DDD = Defined Daily Dose.

## Data Availability

Data are available on request due to ethical restrictions.

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
