# Peer review of "Pharmacoepidemiological Analysis of Antibacterial Agents Used in a Provisional Hospital in Aktobe, Kazakhstan, in the Context of COVID-19: A Comparison with the Pre-Pandemic Period"

_antibiotics, 2023, doi:10.3390/antibiotics12111596_

Round 1
Reviewer 1 Report
Comments and Suggestions for Authors
The manuscript presents interesting comparison between data from a period pre-pandemic and in the pandemic in Aktobe (as it has been reported in many other places around the world). The presentation is clear, concise, and adequate. However, poor comparison is with other studies in other sites around the world. That could improve reader's interest.
Results among your study and other sites should be presented after those in Aktobe.
Discussion could be enriched with precise values. But also, with the local characteristics related to the detected changes in prescription (information, drug availability, trust of physicians and patients in a group of drugs).
Conclusions should be edited to clearly be linked and supported by results.
Author Response
- An extensive discussion was added to compare the results of the city of Aktobe with other studies conducted in other countries.
- Exact values have been added to the discussion. And also, information was added with local features related to the identified changes in prescribing (awareness, availability of the drug, trust of doctors and patients in the group of drugs).
- The conclusions were edited and supported by the results.
Reviewer 2 Report
Comments and Suggestions for Authors
I have reviewed the manuscript entitled “Pharmacoepidemiological analysis of antibacterial agents used in a provisional hospital in Aktobe, Kazakhstan, in the context of COVID-19: a comparison with the pre-pandemic period” for possible publication in “Antibiotics/MDPI”. The content of manuscript represents the interest of researchers as well as clinicians. However, there are still some gaps which must needs to be addressed before proceeding it further. The authors haven’t provided the data for the reason of antibiotic usage, without this information the conclusion will be misleading. I recommend the authors to add more data for type of infections, bacteria etc and the specific reasons for using the antibiotics. For sure the reason of using antibiotics was not only the COVID-19 status.
My specific general comments are:
1. The abstract should be start with a background sentences, then further continue with the aims.
2. The authors should put some results with numbers in the abstract.
3. The conclusion in the abstract needs to be revise.
4. The introduction section is too long, the authors are suggested to remove the unnecessary literature. e.g., paragraph 3, 4, 6, 7, these all are extra literature which is already reported in many studies.
5. Line 90: no need to our full form of WHO as it is already mentioned at line 27.
6. I suggest the authors to keep the introduction section maximum in 4-5 paragraphs.
7. Line 122: The authors are suggested to use the words like present study/current etc., not to use me/our/ etc.
8. The results are with less data. More data is needed to represent the correct usage of antibiotics.
9. After the authors add more data, then the discussion section will be more interesting.
10. Methods are fine, but they need to see for other possibilities.
11. The conclusion section should be written in one paragraph.
Author Response
- Data were added on the reason for the use of antibiotics, data on the type of infections, bacteria, etc., as well as on specific reasons for the use of antibiotics.
- The abstract was started with background suggestions, namely relevance "
- Some results with numbers in the annotation have been added to the annotation.
- The output in the annotation has been changed - The introductory part was shortened
- References 3, 4, 6, 7 have been deleted - Line 90: The full WHO form has been deleted
- We tried to reduce the introductory part to 4-5 paragraphs, but as a result it becomes less than 4000 words
- Personal pronouns have been changed - The final part is written in one paragraph.
Reviewer 3 Report
Comments and Suggestions for Authors
GENERAL COMMENTS
The terms antibacterial, antibiotic and antimicrobial are used interchangeably in the document. Use 'antibacterial' throughout the document.
Eliminate the use of personal pronouns in the document. This may require rephrasing of some sentences/paragraphs
Italicize words as appropriate - in vitro, scientific names, etc
SPECIFIC COMMENTS
These are annotated in the attached PDF document.
Address them as appropriate

Comments on the Quality of English LanguageThe English language used is appropriate for scientific writing.
However, several sections require rephrasing and replacement of words/phrases as indicated in the attached annotated PDF document.
Author Response
- Throughout the document, the words "antibiotics" and "antimicrobial" were changed to "antibacterial"
- Personal pronouns were excluded. Some sentences/paragraphs have been rephrased - Scientific names are in italics
- All the comments that are indicated in the PDF document have been corrected
Round 2
Reviewer 2 Report
Comments and Suggestions for Authors
The authors have done great efforts to revise the manuscript and addressed most of my concerns. However, the manuscript still needs some correction to do before proceeding it further for the possible publication process. My specific comments for the manuscript are:
1. The manuscript should be revised for English proofreading by a native speaker or otherwise authors can use the paid English editing services.
2. Line 13: The authors are recommended to use full forms of abbreviations at their first appearances.
3. Line 25-27: The sentence should be revise and divided into two sentences.
4. Line 44: This is not the correct way to write abbreviation first and then to write full forms in the brackets. The authors are recommended to use full forms of abbreviations at their first appearances.
5. The introduction section is still quite lengthy. If the problem is word count, I suggest the authors to add more details in material & methods, results and discussion section.
6. Line 331: I suggest the authors to add months of the respective years as well.
7. Line 348: sample as comment 4.
8. I suggest the authors to move 4.3 to 4.2 and then change 4.2 as 4.3.
9. The separate statistical section will be more better.
10. I suggest the authors to revise figure 2 as double line bar graph and then add p-values in both figures.
11. I haven’t seen the statistical corelation tests anywhere. Like chi square, t test etc. I suggest the authors to do add some statistical analysis to see the differences and to report the significance.
Comments on the Quality of English LanguageThe manuscript should be revised for English proofreading by a native speaker or otherwise, authors can use the paid English editing services.
Author Response
1. The manuscript has been finalized by a native speaker. The authors tried to correct English to the proper level. Please do not judge strictly.
2. The authors used the full forms of abbreviations when they first appeared.
3. The sentence (Lines 25-27) is divided into two sentences.
4. The authors have corrected abbreviation.
5. The introductory part has been shortened. Some of the information has been moved to the Materials and Methods section.
6. 4.3 was moved to 4.2 7. Statistical analysis was carried out using the t-test method.
Thank you for your recommendations! We are trying very hard to fix them.